# 🖼️ OmniBench: Towards The Future of Universal Omni-Language Models

## Abstract

Recent advancements in multimodal large language models (MLLMs) have aimed to integrate and interpret data across diverse modalities. However, the capacity of these models to concurrently process and reason about multiple modalities remains underexplored, partly due to the lack of comprehensive modality-wise benchmarks. We introduce **OmniBench**, a novel benchmark designed to rigorously evaluate models' ability to recognize, interpret, and reason across **visual**, **acoustic**, and **textual** inputs simultaneously. We define language models capable of such tri-modal processing as the omni-language models (OLMs). OmniBench is distinguished by high-quality human annotations, ensuring that accurate responses require integrated understanding and reasoning across all three modalities. Our main findings reveal that: *i)* open-source OLMs exhibit critical limitations in instruction-following and reasoning capabilities within tri-modal contexts; and *ii)* most baselines models perform poorly (below 50% accuracy) even when provided with alternative textual representations of images or/and audio. These results suggest that the ability to construct a consistent context from text, image, and audio is often overlooked in existing MLLM training paradigms. To address this gap, we curate an instruction tuning dataset of 84.5K training samples, **OmniInstruct**, for training OLMs to adapt to multimodal contexts. We advocate for future research to focus on developing more robust tri-modal integration techniques and training strategies to enhance OLM performance across diverse scenarios. Codes and data could be found at our repo.

## 1 Introduction

The rapid advancement of artificial intelligence has ushered in a new era of multimodal large language models (MLLMs), capable of processing and interpreting diverse data types mainly involving images, audio, and text (Li & Lu, 2024). These models aim to emulate human-like understanding of the world by integrating information across multiple sensory modalities and learning a comprehensive context from the environment. While significant strides have been made in developing MLLMs that can handle two of the modalities, the ability to concurrently process and reason about the three aforementioned modalities remains a frontier yet to be fully explored.

The social impact of these MLLMs is far-reaching, providing transformative capabilities for a variety of domains. In healthcare, VLMs and ALMs have contributed to diagnosing (Liu et al., 2023a; Hemdan et al., 2023), and potentially combining *three modalities* (Meskó, 2023). The integration of all vision, audio and text modalities is expected to significantly improve diagnostic accuracy and patient interaction, making healthcare more accessible and efficient. In urban environments, ALM can contribute to improving safety and traffic management by incorporating urban sound event detection during autonomous driving, such as recognizing audio of emergency vehicles and recognize their types or location with supplementary visual modality (Sun et al., 2021). In addition, audio contributes to biodiversity monitoring (Terenzi et al., 2021; Liang et al., 2024a) and can be greatly enhanced by MLLM's ability to analyse both audio and video from a variety of sensors. Finally, it may help robotics or LLM agents to provide better human-computer/robotic interaction (HCI/HRI) service in day-to-day life (Liang et al., 2024b; Su et al., 2023).

The challenge in advancing MLLMs lies not only in their development but also in our capacity to evaluate their performance comprehensively. Current benchmarks often solely focus on image or

audios, or limited image-text (Yue et al., 2024; Zhang et al., 2024) or audio-text combinations (Yang et al., 2024) for the dual-modality vision-language models (VLMs) (Laurençon et al., 2024) or audio-language models (ALMs) (Chu et al., 2023a; Deng et al., 2023). This gap in evaluation tools has hindered the community to assess and improve the holistic capabilities of models right before the dawn of general-purpose MLLMs.

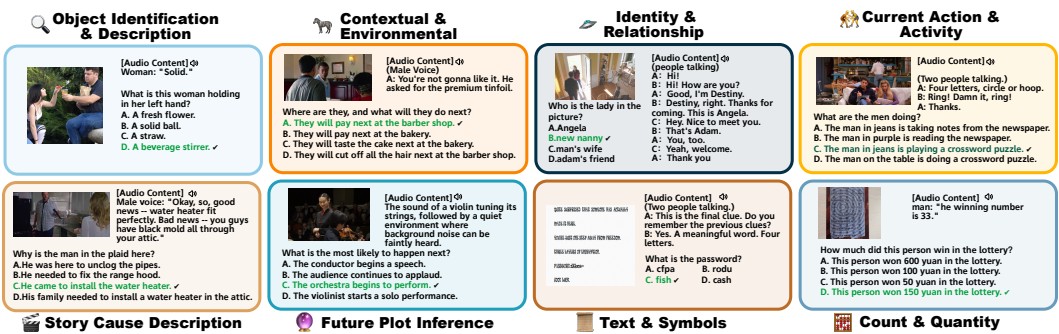

Figure 1: Example Data from Different Categories. The main contextual information is provided by the corresponding image and audio, while the question and options are expressed with text. Playable audio demos are available at the demo page.

To address this critical need, we introduce **OmniBench**, a pioneering universal multimodal benchmark designed to rigorously evaluate MLLMs' capability to recognize, interpret, and reason across visual, acoustic, and textual inputs *simultaneously*, which we define as the *omni-understanding and reasoning* ability of the omni-language models (**OLM**s) (Sun et al., 2024; Zhan et al., 2024; Lu et al., 2024b). For instance, one can only derive the correct answer of the sample question in Figure 1 by: *1*) recognizing elements from the given image and audio to reconstruct the context; *2*) interpreting the semantics and relationships among the multimodal objects according to the textual instruction formed as question and options; *3*) reasoning and then answering with the complementary information from all the modalities. We distinguishes OmniBench by enforcing a unique constraint: accurate responses necessitate an integrated understanding and reasoning of *all multimodal contexts*. This approach ensures a more realistic and challenging assessment of multimodal large language models, mirroring the complex, interconnected nature of human cognition. To ensure the evaluation reliability, the development of OmniBench relies on high-quality human annotations. Furthermore, OmniBench additionally includes the answer rationales provided by the annotators to enhance the validity and ensure the benchmark aligned with human-level understanding.

Our initial findings using OmniBench reveal critical limitations in the omni-understanding capabilities of existing MLLMs:

- Although the existing open-source omni-language models have been trained with data in the three modalities, most of them can surpass the performance of random guess accuracy but sometimes hard to follow the instruction when provided with image and audio together in certain cases.

- Compared to the open-source OLMs, the proprietary models perform better overall but suffer from more accuracy drops when ablating the image or audio input.

- In the context of using text as an alternative source of information to corresponding audio and images, the open-source VLMs and ALMs show relatively better results but remain in a preliminary level of capability to understand the given tri-modality context.

These results underscore the importance of OmniBench as a tool for identifying areas of improvement and guiding research in multimodal systems. In the following sections, we 1) detail the data collection protocol of OmniBench; 2) present our evaluation results on current state-of-the-art MLLMs; 3) introduce the **OmniInstruct** dataset for omni-language model supervised fine-tuning; and 4) discuss the implications of our findings for the future of research and development. Through OmniBench, we aim to catalyze advancements in MLLMs, pushing the boundaries of artificial intelligence towards true omni-understanding capabilities.

## 2 RELATED WORK

**Multimodal Large Language Models.** Recent advancements in multimodal large language models (MLLMs) have aimed to integrate and interpret data across diverse modalities. In the audio domain, models like Whisper (Radford et al., 2022), BEATs (Chen et al., 2022), MERT (Li et al., 2023b), and CLAP (Wu et al., 2023b) have been developed as specialized encoders for speech, general audio, acoustic music, and music-text, respectively. These have been incorporated into more comprehensive systems such as SALMONN (Tang et al., 2023), LTU (Gong et al., 2023), Mu-llama, MusiLingo, and Audio-Flamingo. Notable progress in audio perception and instruction-following includes SALMONN (Tang et al., 2023), BLSPN (Wang et al., 2023a), Speech-LLaMAN (Wu et al., 2023a), and Qwen-Audio (Chu et al., 2023b), all demonstrating promising capabilities in audio-focused dialogues. In the visual domain, large visual language models have made significant strides, often leveraging pre-trained image encoders (Dosovitskiy, 2020; Touvron et al., 2020; Liu et al., 2021; Radford et al., 2021; Zhai et al., 2023). Notable examples include BLIP2 (Li et al., 2023a), which uses a Q-Former for visual-textual alignment, LLaVA (Liu et al., 2024b), which employs GPT-4 generated instruction data, and its successor LLaVA-Next (Liu et al., 2024a). Building on the LLaVA framework, models like QwenVL (Bai et al., 2023), CogVLM (Wang et al., 2023b), and Yi-VL (Young et al., 2024) have achieved significant success through extensive pre-training. Despite these advancements, most existing MLLMs focus on a single modality for input processing while generating textual responses. While some models can process textual, audio, and visual inputs simultaneously, open-source models in this field generally exhibit less competitive capabilities compared to their closed-source counterparts. In this context, we define omni-language models (**OLM**s) as those capable of processing at least three different modalities of data simultaneously[1].

**Multimodal Understanding Benchmark.** The vision-language benchmarks aim to test models' ability to combine visual and language data in tasks like OCR, spatial awareness, multimodal information retrieval (e.g. SciMMIR (Wu et al., 2024a)), and multimodal reasoning skills. MM-Vet (Yu et al., 2023) focuses on visual question answering (VQA), requiring models to interpret visual data and respond to queries. MMBench (Liu et al., 2023c) evaluates models via multiple-choice tasks in both Chinese and English, covering diverse domains. MMStar (Chen et al., 2024a) conducts multi-task evaluations to test multimodal fusion capabilities. MMMU (Yue et al., 2024) and CM-MMU (Zhang et al., 2024) assess model performance on complex vision-language tasks, emphasizing sophisticated multimodal reasoning. MMRA (Wu et al., 2024b) is designed to evaluate the models' multi-image relational association capability. In addition, there are several audio-understanding benchmarks. Aishell1 (Bu et al., 2017), Aishell2 (Du et al., 2018), and Librispeech (Panayotov et al., 2015) are designed for automatic speech recognition, while ClothoAQA targets audio QA tasks. For automatic audio captioning and vocal sound classification, researchers have curated Clotho (Drossos et al., 2020) and VocalSound (Gong et al., 2022). However, there is a significant lack of comprehensive understanding benchmarks to assess MLLMs' ability to simultaneously process complementary information from the textual, audio, and visual inputs.

**Audio-Visual Understanding Datasets.** In previous works on audio-visual question answering (AVQA), the focus has predominantly been on identifying visual objects, sounds, and their interrelations to foster multimodal understanding. For instance, the Pano-AVQA dataset (Yun et al., 2021) explores 360-degree panoramic video understanding with 5.4k videos and 51.7k QA pairs. However, it limits its scope to identifying existing objects or locations, omitting questions on causal inference and abstract concepts. Similarly, the Music-AVQA dataset (Li et al., 2022) comprises 9.3k videos, including 1.9k synthesized entries, and 45.9k QA pairs. These videos typically feature simple scenarios of music performances by one or two players or within a music chamber. The questions focus on existing objects, time and location, counting, and relationships but fail to address special symbols like music score images or causal inference. The Music-AVQA-2.0 dataset (Liu et al., 2024c) enhances Music-AVQA by collecting 1,230 manually curated musical ensemble performance videos and 8.1k newly created QA pairs designed to complement and diversify the original dataset, addressing biases in annotation and instrument types. However, it maintains the original types of QA pairs, not expanding into new categories of questions. The AVQA dataset (Yang et al., 2022), contains 57k QA pairs that do not necessarily require integration of both modalities for answering, illustrating its

---

[1]We target the models able to concurrently process image, audio, and text as a starting point since these are the most well-explored modalities in the field, but the "omni" concept is extendable.

limitation in truly multimodal inquiry. For example, the presence of an auditory signal like a train whistle isn't essential to deduce an action such as the lowering of a gear lever at a railroad crossing, suggesting that answers could be surmised from visual cues alone. This dataset includes questions on time, location, existing objects, causality, purpose, and counting, yet lacks coverage of actions, symbol concepts, and associations. VALOR dataset (Chen et al., 2023) is an audiovisual-language dataset designed for tri-modality model pre-training, comprising 1.18 M videos sourced and curated from AudioSet (Gemmeke et al., 2017). Recognizing captions derived from ASR or alt-texts fail to adequately align audio-language modalities; VALOR annotates audio-visual content by humans based on AudioSet tags to establish a clear correspondence among 3 modalities. VALOR is good for pre-training but does not include instruction following QA pairs. These gaps in current datasets underscore the need for more comprehensive AVQA datasets and evaluation benchmarks that can challenge and accurately measure a model's capacity for deep multimodal integration and abstract reasoning, which are critical for advancing multimodal understanding in AI.

## 3 OMNIBENCH

The OmniBench aims to create the first comprehensive benchmark for evaluating multimodal large language models that support simultaneous image, audio, and text inputs. While OmniBench is designed to evaluate the understanding capability of MLLMs on cross-modality complementary information, the models are required to interpret the multimodal input and provide accurate text answer. The problem could be formulated as following: given a tuple of (image, audio, text), the model is required to recognize the objects, re-build the contexts, and conduct reasoning based on the given information. The design logic and statistics of the dataset and the annotation protocols are introduced in this section.

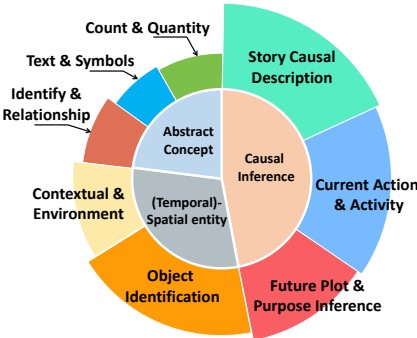

Figure 2: The Taxonomy and Proportions of the 1142 Test Samples in OmniBench. The inner circle refers to the three major categories, and the traffic circle refers to the fine-grained task types.

### 3.1 BENCHMARK DESIGN

Building on the foundation of existing multimodal benchmarks, our OmniBench introduces a refined taxonomy for task categorization that effectively captures a wide range of cognitive and reasoning abilities. As demonstrate in Figure 2, our framework organizes tasks into three primary categories: (1) *(temporal)-spatial entity*, which includes *Object Identification* for recognizing distinct entities and *Contextual & Environmental* for discerning the setting or backdrop of the events; (2) *causal inference*, comprised of *Story Cause Description* to infer narrative drivers, *Current Action & Activity* to understand ongoing dynamics, and *Future Plot and Purpose Inference* to anticipate subsequent developments; and (3) *abstract concept*, involving *Identity & Relationship* to identify and relate entities, *Text & Symbols* for symbolic interpretation, and *Count & Quantity* for numerical reasoning. This taxonomy is designed to evaluate both foundational perceptual skills and complex cognitive processes, thereby providing a comprehensive assessment of multimodal language models' (MLLMs) abilities to integrate and interpret diverse information sources. OmniBench includes **1142** question-answer pairs, with details on task types, text length, and the characteristics of images and audio presented in Table 1. The audio content of the dataset is categorized into speech, sound

Table 1: The Statistics of OmniBench Across Task Types. The word lengths of four options for each question are first averaged, and then the averages are calculated in group.

| Statistic | Causal Inference | | | (Temporal-)Spatial Entity | | Abstract Concept | | | |
|---|---|---|---|---|---|---|---|---|---|
| Sub-class of QA | Current Action & Activity | Story Description | Plot Inference | Object Identification & Description | Contextual & Environmental | Identity & Relationship | Text & Symbols | Count & Quantity | Overall |
| *Quantity* | | | | | | | | | |
| **Total** | 251 | 230 | 237 | 211 | 141 | 32 | 25 | 15 | 1142 |
| **Speech** | 78 | 182 | 179 | 162 | 104 | 31 | 22 | 13 | 771 |
| **Sound Event** | 147 | 27 | 37 | 28 | 25 | 1 | - | - | 265 |
| **Music** | 26 | 21 | 21 | 21 | 12 | - | 3 | 2 | 106 |
| *Word Length* | | | | | | | | | |
| **Question** | 4.68 | 5.75 | 7.47 | 7.00 | 6.85 | 6.22 | 7.32 | 8.72 | 6.25 |
| **Option** | 8.85 | 7.77 | 8.92 | 6.47 | 5.68 | 10.38 | 11.22 | 6.60 | 8.81 |
| **Img. Rationale** | 18.27 | 19.62 | 24.40 | 24.94 | 18.34 | 22.69 | 24.80 | 29.16 | 21.19 |
| **Audio Rationale** | 23.11 | 20.50 | 24.40 | 20.97 | 18.27 | 24.92 | 23.10 | 53.84 | 22.90 |
| **Audio Content** | 13.21 | 17.91 | 29.87 | 28.03 | 14.41 | 19.01 | 23.31 | 35.16 | 18.37 |
| *Multimodal Info.* | | | | | | | | | |
| **Img. Width** | 1283.75 | 1291.60 | 2394.93 | 1430.03 | 1141.39 | 1395.53 | 1338.51 | 1787.36 | 1322.36 |
| **Img. Height** | 842.32 | 776.11 | 2089.93 | 799.47 | 728.06 | 840.15 | 761.58 | 1168.04 | 818.64 |
| **Audio Len.** (s) | 7.35 | 9.82 | 11.22 | 11.43 | 8.03 | 8.63 | 11.43 | 15.63 | 9.22 |

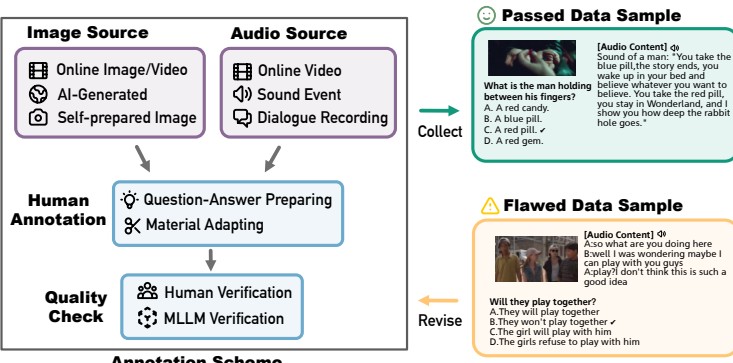

Figure 3: OmniBench Annotation Scheme. The annotation example shows flawed data that does not pass inspection because the information in audio *alone* is sufficient to answer. The audio in the flawed sample will then be sent back to annotators to edit.

events, and music, enriching the diversity of stimuli for evaluating the models' tri-modal capabilities and aiding in the development of future omni-language models.

## 3.2 ANNOTATION PROTOCOL

**Annotation Scheme.** Our annotation scheme is built upon a fundamental principle: the correct answer to each question must require information from both the image and audio components. This ensures that the benchmark effectively evaluates the model's ability to analyze information across modalities. As shown in Figure 3, we implemented a rigorous annotation pipeline consisting of three stages: initial annotation, human inspection, and model inspection. Data samples that failed to meet our criteria at any stage were returned to annotators for revision, ensuring high-quality, multimodal-dependent samples. Through the whole process, 16 *annotators and* 5 *quality inspectors* are involved, all are full-time industrial data annotation employee with higher education backgrounds.

The questions are formalized as multi-choice question-answering (MCQ) but try to maintain a consistent logic that suggests the only one possible and accurate answer, *i.e.*, they can be potentially further re-organized into blank filling questions. Furthermore, when constructing the options, the annotators need to ensure at least one confusing wrong option. To ensure question difficulty, annotators were required to verify that questions and options were not trivially easy, lacked distinguishable patterns, and could not be answered by state-of-the-art MLLMs using image information alone. GPT-4 are allowed to use to provide initial annotator self-assessments of question quality.

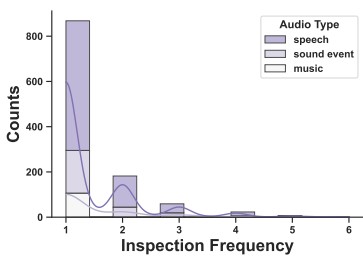

Figure 4: The Distribution of Inspection Frequency of the Passed Samples in OmniBench.

We restrict the images with a minimum resolution of 480P (854x480 pixels) and audio clips with a maximum duration of 30 seconds.

We implemented strict measures to maintain diversity across the dataset. This includes varying image and audio sources, limiting the frequency of individual speakers in audio clips to no more than five occurrences, and restricting the replication of similar instructions or questions. For instance, questions about specific environmental contexts were limited up to three samples. Importantly, annotators were required to provide **rationales** for correct answers, detailing the specific information that should be derived from the image and audio modalities respectively. This approach not only aided in quality inspection but also laid the groundwork for future fine-grained evaluation.

**Quality Control.** Our quality control process was two-fold, including human inspection round and automatic inspection round assisted by MLLM. First, all annotated instruction-response pairs undergo cross-inspection by human annotators. Inspectors provide detailed reasons for any samples that failed to meet our stringent criteria, allowing for targeted revisions. Samples that pass human inspection are then subjected to a secondary check using a vision-language model LLaVA-1.6-34B (Liu et al., 2024a), where the the automatic quality inspection model is selected by considering trade-off between efficiency and performance. This automated process evaluates each sample under various ablation settings: *image and text only*, *audio transcript and text only*, and *text only* (repeated three times). Samples are only accepted if the model either rejected the task or made mistakes under these limited-information scenarios, confirming the necessity of both visual and auditory information for correct responses. We plot the distribution of the inspection frequency of the passed samples in Figure 4, where we could find that 76% (868) of the passed samples do not require further modification under a well-defined annotation framework and 21.1% of them requiring 1-2 times of revision. During the iterative quality checking, 9.58% (121) QA pairs are defined as "hard to recycle" and dumped after revisions and discussions.

### 3.3 OMNIINSTRUCT

To improve the model capability of tri-modal reasoning, we develop the **OmniInstruct** dataset to facilitate the supervised fine-tuning of models. This dataset leverages the following data sources: the MSRVTT-QA (Xu et al., 2017), AVQA (Yang et al., 2022) and Music-AVQA2.0(Liu et al., 2024c), all of which contain visual, audio and corresponding QA text resources. MSRVTT-QA and AVQA consist of short video clips, typically ranging from 10 to 20 seconds, with minimal scene changes, and music-AVQA 2.0 dataset include 1 minute music performance video. We only adopt the train and validation split of this dataset and regard the whole OmniBench as the test set of the task.

To construct a dataset that aligns with the challenges proposed in OmniBench, we enhanced each question to connect with an audio track and an image extracted from the corresponding video and filter it with VLMs for better quality. Notably, we avoid the first and last five frames of each video to exclude transitional or obscure incomplete scenes that might distort the task's focus. For MSRVTT-QA train and valid subset, we discard videos without audio tracks and retain a dataset comprised 6,176 videos from the original set that include audio tracks alongside 151.7k QA pairs directly related to these videos. Then we use InternVL-2-76B to filter the questions from the three datasets mentioned above to delete (1) questions that can be answered only with an image, (2) questions irrelevant with image, potentially answerable only with audio, and (3) ambiguous or non-logical

questions, where the detailed prompt and statistics could be found at Figure 6 and Table 7, Appendix B. Only 93k data samples remain for training and validation.

This curated dataset is essential for evaluating the nuanced capabilities of multimodal large language models to interpret and integrate multiple types of information, which is a first step towards enhancing reasoning performance and applicability in omni-modality scenarios.

# 4 EXPERIMENT SETTINGS

**Baseline Systems**    We select three groups of MLLM baselines according to the modalities available: *(i) omni-language models*: MIO-Instruct (Wang et al., 2024b), AnyGPT (Zhan et al., 2024), Video-SALMONN (Sun et al., 2024), UnifiedIO2 series (Lu et al., 2024b); *(ii) vision-language models*: InternVL-2 series (Chen et al., 2024b), Qwen2-VL series (Wang et al., 2024a), Deepseek-VL (Lu et al., 2024a), LLaVA-One-Vision series (Li et al., 2024), Cambrian series (Tong et al., 2024), Xcomposer2-4KHD (Dong et al., 2024), Idefics2 (Laurençon et al., 2024) as well as the derived Mantis-Idefics2 (Jiang et al., 2024); *(iii) audio-language models*: LTU series (Gong et al., 2023), Mu-LLaMA (Liu et al., 2023b), MusiLingo (Deng et al., 2023), Qwen-Audio series (Chu et al., 2023a), SALMONN-Audio (Sun et al., 2024) and Audio-Flamingo (Kong et al., 2024). We also include the API calls from proprietary models that could support image-text or audio-text inputs, including GPT4-o, Gemini Pro, Reka and Claude-3.5-Sonnet (Achiam et al., 2023; Team et al., 2023; Ormazabal et al., 2024; Anthropic, 2024). We do not conclude them as in the group of VLMs, ALMs or OLMs (even not a single model) in our context at the moment since the mechanisms behind these models are not revealed[2]. Besides, we invite three musician with higher education background to test on the benchmark and use the average accuracy as a human expert baseline.

**Omni-Understanding Evaluation.**    The main focus of OmniBench is to evaluate how well could the MLLMs understand and reconstruct the context given information from image ($I$), audio ($A$) and text ($T$) modalities. Setting up questions with four available options for the models, we use accuracy, *i.e.*, the ratio matched letter of the correct option and model response, as the evaluation metric (*n.b.*, the accuracy of a random guess model is $25\%$ under this setting). Additionally, we test the models in an ablation setting of removing one of the image or audio inputs to further reveal a more comprehensive reasoning capability of the baselines and verify the robustness of our benchmark. For baseline systems, please refer to section 4.

**Textual Approximation of Image and Audio.**    For most of the existing MLLMs that only support two input modalities (($I, T$) or ($A, T$)), we build up a simulated evaluation setting allowing us to explore the potential of these models to become omni-language models in the future. We use the audio transcript ($A'$) annotated by human as the alternative of the audios to enable the evaluation on vision-language models. Regarding the audio-language models, we generate high-quality detailed captions of images ($I'$) automatically with a state-of-the-art VLM, InternVL-2-76B. In such an approximated evaluation setting, models go through the same process of inference and metric calculation as the vanilla one with textual alternatives of images or audios.

# 5 FINDINGS

## 5.1 RESULTS ON OMNI-LANGUAGE MODELS

Table 2 demonstrates that open-source omni-language model (OLM) baselines surpass random guessing accuracy across various settings. Notably, the UnifiedIO2 series demonstrates inconsistent performance scaling with model size, indicating challenges in effectively leveraging increased capacity for multimodal understanding but still much lower than human experts ($63.19\%$ accuracy with a Fleiss' Kappa value of $0.421$ as inter-annotator agreement).

Despite overall poor performance, open-source baselines generally exhibit higher accuracy on speech audio, indicating a potential bias towards speech data. In contrast, Gemini-1.5-Pro and Reka-core-20240501, the two available proprietary models evaluated in this tri-modal setting, shows more promising results. Regarding the scores across audio types, the Gemini-1.5-Pro shows a more balanced performances while Reka-core-20240501 showing a lag on modeling the sound events. Besides, Video-Salmonn, developed by Bytedance, and Gemini, developed by Google, provide better results on music subsets compared to their performance on speech and music., potentially due

---

[2]The authors conclude from an investigation on September 22, 2024.

Table 2: Overall Omni-Undesratnding Results on Baseline Omni-Language Models. The overall (Image & Audio), image-ablated and audio-ablated results on all samples are provided.

| Input Context | Image & Audio | Audio | Image |
|---|---|---|---|
| AnyGPT (7B) | 18.04% | 16.20% | 20.05% |
| video-SALMONN (13B) | 35.64% | **35.90%** | **34.94%** |
| UnifiedIO2-large (1.1B) | 27.06% | 29.07% | 29.07% |
| UnifiedIO2-xlarge (3.2B) | 38.00% | 31.17% | 34.76% |
| UnifiedIO2-xxlarge (6.8B) | 33.98% | 32.49% | 33.45% |
| Gemini-1.5-Pro | **42.91%** | 27.93% | 26.09% |
| Reka-core-20240501 | 30.39% | 23.12% | 30.65% |
| Human Expert | **63.19**% | - | - |

Table 3: OLM Baselines Overall Results Grouped by Audio Type and Task Category. The accuracy numbers calculated by different audio types are at the upper table and the accuracy accross tasks categories are placed at the bottom table.

| Model | Speech | Sound Event | Music |
|---|---|---|---|
| AnyGPT (7B) | 17.77% | 20.75% | 13.21% |
| Video-SALMONN (13B) | 34.11% | 31.70% | **56.60%** |
| UnifiedIO2-large (1.1B) | 25.94% | 29.06% | 30.19% |
| UnifiedIO2-xlarge (3.2B) | 39.56% | 36.98% | 29.25% |
| UnifiedIO2-xxlarge (6.8B) | 34.24% | 36.98% | 24.53% |
| Gemini-1.5-Pro | **42.67%** | **42.26%** | 46.23% |
| Reka-core-20240501 | 31.52% | 26.04% | 33.02% |

| Accuracy ↑ | Causal Inference | | | (Temporal-)Spatial Entity | | Abstract Concept | | |
|---|---|---|---|---|---|---|---|---|
| Sub-class of QA | Action & Activity | Story Description | Plot Inference | Object Identification & Description | Contextual & Environmental | Identity & Relationship | Text & Symbols | Count & Quantity |
| AnyGPT (7B) | 19.52% | 16.52% | 14.77% | 22.27% | 15.60% | 21.88% | 12.00% | 33.33% |
| Video-SALMONN (13B) | 31.47% | 28.26% | 25.74% | 62.56% | 36.88% | **37.50%** | 20.00% | 6.67% |
| UnifiedIO2-large (1.1B) | 29.88% | 20.87% | 31.65% | 30.81% | 23.40% | 18.75% | 24.00% | 6.67% |
| UnifiedIO2-xlarge (3.2B) | 32.27% | **33.48%** | 31.65% | **63.03%** | 34.04% | 34.38% | 24.00% | 20.00% |
| UnifiedIO2-xxlarge (6.8B) | 32.27% | 29.13% | 29.96% | 48.82% | 34.75% | 25.00% | 8.00% | **46.67%** |
| Gemini-1.5-Pro | **41.83%** | 30.87% | **32.91%** | 62.56% | **60.28%** | 31.25% | **28.00%** | 13.33% |
| Reka-core-20240501 | 25.50% | 24.78% | 20.68% | 49.76% | 39.01% | 28.12% | **28.00%** | 6.67% |

to their large corpus of music videos, though the music ethics of training foundation models are still underdiscussion (Ma et al., 2024). Moreover, the comparison of Gemini-1.5-Pro's performance across full input context and ablated settings (image-removed and audio-removed) suggests that it effectively leverages information from all modalities to enhance its reasoning capabilities. While it demonstrates superior overall performance and balanced accuracy across audio types compared to open-source alternatives, its accuracy remains below 50%.

These findings underscore the challenging nature of OmniBench and highlight substantial room for improvement in multi-modal reasoning tasks. We anticipate the development of more competitive models on our benchmark in the near future, which will further advance the field of multi-modal AI.

**Breakdown Results.** We present the breakdown of the performance of open-source omni-language model baselines across different audio types and task categories in the OmniBench evaluation. The results reveal inconsistent performance patterns across audio types, with some models showing higher accuracy on sound events or music compared to speech. Across task categories, models tend to perform better on object identification and description tasks, while struggling with more reasoning tasks such as plot inference and story description, as illustrated by Table 3. This might be because visual entity recognition is an essential component for image captioning and other type of pre-training dataset. We observe Gemini provides significantly better results on the context/environment entities other than object entities. Furthermore, most of the models perform really bad on quantity & counting tasks. But scaling up of UnifiedIO model contributes a lot to this type of task. And scaling up from 1.1B to 3.2B benefits all senarios.

These findings highlight current limitations of OLMs in integrating information across modalities.

Table 4: Results on Textual Audio Approximation Experiments. All the audios are represented in text transcript. The results are divided into groups of vision-language models and omni-models. We use the text transcript to approximate the audios in this setting. Boldface shows the best model performance, and underline shows the best open-source model.

| Input Context | Image & Audio Transcript | Audio Transcript | Image |
|---|---|---|---|
| InternVL-2-2B | 42.29% | 27.32% | 28.11% |
| InternVL-2-8B | 50.79% | 33.63% | 33.36% |
| InternVL-2-26B | 51.75% | 31.87% | 33.89% |
| InternVL-2-40B | 54.29% | 31.96% | 34.76% |
| Qwen2-VL-Chat-2B | 42.47% | 31.44% | 38.09% |
| Qwen2-VL-Chat-7B | 48.60% | 32.05% | 36.87% |
| Deepseek-VL-Chat-7B | 39.67% | 29.51% | 26.27% |
| Idefics2-8B | 45.10% | 32.31% | 34.41% |
| Mantis-Idefics-8B | 46.15% | 36.43% | 32.57% |
| LLaVA-OneVision-0.5B | 38.00% | 31.79% | 31.17% |
| LLaVA-OneVision-7B | 47.02% | 31.70% | 29.68% |
| Cambrian-8B | 42.12% | 31.35% | 32.22% |
| Cambrian-13B | 45.01% | 31.96% | 33.98% |
| Cambrian-34B | 46.76% | 30.12% | 33.01% |
| XComposer2-4KHD (7B) | 43.96% | 29.25% | 30.65% |
| GPT4-o (0513) | 57.62% | 45.71% | 42.21% |
| GPT4-o (0806) | 51.14% | 47.55% | 31.44% |
| GPT-4-o-mini | 49.04% | 39.23% | 34.06% |
| Gemini-1.5-Pro | 44.40% | 22.50% | 26.09% |
| Reka-core-20240501 | 46.58% | 34.59% | 30.65% |
| Claude-3.5-Sonnet | **59.37%** | 33.54% | **43.08%** |
| GPT-4V-Preview | 38.18% | 41.24% | 25.57% |
| GPT-4V-0409 | 33.36% | **45.80%** | 32.75% |
| UnifiedIO2-large (1.1B) | 34.33% | 31.96% | 29.07% |
| UnifiedIO2-xlarge (3.2B) | 43.17% | 34.50% | 34.76% |
| UnifiedIO2-xxlarge (6.8B) | 40.81% | 29.77% | 33.45% |

## 5.2 TEXTUAL APPROXIMATION ON IMAGES AND AUDIOS

As the absence of strong OLM baselines on OmniBench, we further introduce the text alternatives of images ($I'$) and audios ($A'$) to embrace more dual-modal MLLMs to analyze the current research progress in the field. The results using audio transcripts and image captions are put in Table 4, Table 5 correspondingly (results of ($I', A'$) setting at Table 6, Appendix A).

Table 5: Results on Textual Image Approximation Experiments. All the images are represented in text caption. The results are divided into groups of audio-language models and omni-models.

| Accuracy ↑ | All Audio Types | | | Speech | Sound Event | Music |
|---|---|---|---|---|---|---|
| Input Context | Image Caption & Audio | Audio | Image Caption | Image Caption & Audio | | |
| LTU (7B) | 23.29% | 23.91% | 23.12% | 25.42% | 20.00% | 16.04% |
| Mu-LLaMA (7B) | 1.58% | 1.84% | 1.84% | 1.56% | 1.13% | 2.83% |
| MusiLingo-long-v1 | 13.66% | 11.03% | 9.02% | 11.93% | 13.96% | 25.47% |
| Audio-SALMONN (13B) | 34.76% | 32.66% | 33.36% | 34.50% | 29.43% | **50.00%** |
| Qwen-Audio-Chat (7B) | 17.51% | 16.64% | 18.39% | 14.66% | 22.64% | 25.47% |
| Qwen2-Audio-7B-Instruct | **40.72%** | **35.20%** | **35.29%** | **40.60%** | **41.89%** | 38.68% |
| Audio-Flamingo (1.3B) | 24.78% | 23.82% | 24.78% | 26.98% | 21.51% | 16.98% |
| Gemini-1.5-Pro | 38.62% | 28.02% | 21.02% | 39.82% | 33.96% | 41.51% |
| Reka-core-20240501 | 29.42% | 23.12% | 26.27% | 28.53% | 29.43% | 35.85% |
| UnifiedIO2-large (1.1B) | 29.16% | 29.07% | 29.33% | 28.40% | 32.45% | 26.42% |
| UnifiedIO2-xlarge (3.2B) | 32.22% | 31.17% | 30.21% | 32.43% | 32.45% | 30.19% |
| UnifiedIO2-xxlarge (6.8B) | 32.05% | 32.49% | 27.15% | 31.13% | 38.87% | 21.70% |

**Performance Changes of Open OLMs.** We select the UnifiedIO-2 series to conduct the textual approximation experiments due to their relatively robust performances in the vanilla evaluation setting suggested in Table 2. Compared with the vanilla setting, all three UnifiedIO-2 models show performance gains, averagely at $6.42\%$, in the audio replacement setting and average performance drops in the replaced-image ($1.87\%$) and both-repaced settings ($0.12\%$). This indicates the shortcoming of existing OLMs on modeling the audio on the one hand, and the potential noise in the generated image captions compared to the human-written audio transcripts on the other hand.

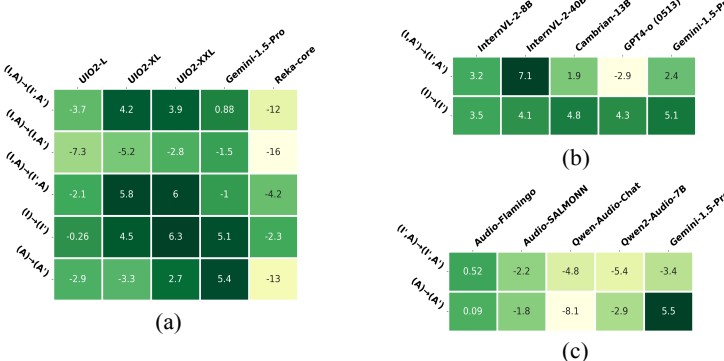

Figure 5: The Performance Changes Brought By Textual Alternatives. The numbers in the cell suggest the accuracy change. *(a)* includes the UnifiedIO2 OLMs and the proprietary models supporting tri-modal inputs. *(b)* and *(c)* consists of VLMs and ALMs grouped with Gemini-1.5-Pro for comparison.

**Performances of Dual-modal MLLMs.** In the setting of using text as the alternatives of audios and images, the VLMs show generally better results than ALMs (Table 4 vs Table 5) even compared with open-source model with similar model size. This could be caused by : 1) more available research resources have been put in VLMs to develop datasets and cross-modality alignment architectures, leading to higher instruction following rate and accuracy compared to ALMs; 2) the audio data are naturally harder (and hence more expensive) to annotate; and 3) audio typically has longer sequence tokens and requires more computational resource compared to text and image, making it harder to scale up. If $I'$ and $A'$ have the information loss ratio when converted from $I$ and $A$, it seems to be easier for the researchers to train the future omni-language models from exisiting VLMs rather than ALMs. Besides, we can observe Claude-3.5 and GPT-4o are generally the best two VLMs, significantly better compared to open-source VLMs. And Qwen2-audio and Gemini are the best two ALMs in speech and audio, and Audio-SALMONN is the best on music. Moreover, we can see significant difference on different type of audio, i.e., LTU and audio-flamingo are worse for music compare to speech and audio, while Qwen-audio which include music on pre-training provides better results on music compared to speech. And MusiLingo only use music for pre-training perform worse in speech and sudio.

**Pure Textual Evaluation.** The performance gaps brought by the replaced textual image and audio descriptions are in revealed in Figure 5. Notably, the majority of models demonstrate improved accuracy when processing textual representations of multimodal data compared to their performance on either image captions or audio transcripts alone. This suggests that these models show stronger reasoning capability when equipped with information from multiple textual sources rather than handling raw multimodal inputs. For instance, Qwen2-Audio-7B-Instruct shows a significant jump in accuracy from 39.05% (audio transcript only) and 39.67% (image caption only) to 47.02% when given both textual representations. Similarly, proprietary models like GPT4-o and Claude-3.5-Sonnet exhibit substantial gains, with GPT4-o (0513) achieving an impressive 60.60% accuracy in the pure textual setting.

## 6 CONCLUSION AND FUTURE STUDY

The proposed novel multimodal benchmark, OmniBench, reveals that current open-source multimodal large language models struggle with simultaneous processing of visual, acoustic, and textual inputs. We observed a general bias towards speech audio and superior performance of vision-language models over audio-language models when using textual approximations. These findings underscore the need for more appropriate architecture designs for multimodal integration, diverse datasets for training, and techniques to reduce modality bias. OmniBench serves as a crucial tool for guiding advancements in multimodal language models, driving progress towards more advanced and versatile models towards human-like multimodal understanding and reasoning.

ETHICS STATEMENT

Our research on OmniBench and the development of multimodal language models raises several important ethical considerations:

- Data Collection and Privacy: All image and audio data used in OmniBench was collected from public sources or created specifically for this research. We took care to remove any personally identifiable information. For human-recorded audio, participants provided informed consent and were compensated fairly for their time.
- Potential Biases: We acknowledge that the dataset may contain inherent biases in terms of language, cultural representation, and types of scenarios depicted. We have made efforts to include diverse content, but further work is needed to fully characterize and mitigate these biases. Users of OmniBench should be aware of these limitations.
- Responsible Disclosure: We will release OmniBench publicly to foster open research, but with appropriate use guidelines. The OmniInstruct dataset will be made available to researchers who agree to terms of responsible use.

We are committed to ongoing evaluation of the ethical implications of this work as the field of multimodal AI continues to advance rapidly.

REPRODUCIBILITY STATEMENT

We have made significant efforts to ensure the reproducibility of our work on OmniBench and the associated experiments:

- Dataset: The complete OmniBench dataset, including all images, audio files, and question-answer pairs, will be made publicly available upon publication. Detailed information about the data collection process, annotation guidelines, and quality control measures are provided in section 3 and Appendix B.
- Code: We have developed and will release a comprehensive codebase that includes: Scripts for data preprocessing and formatting; Implementation of all evaluation metrics; Code for running experiments.
- Model Evaluation: For all baseline models evaluated, we provide detailed specifications. For proprietary models, we specify the exact API versions used and the dates of access.
- Reproducibility Challenges: We acknowledge that exact reproduction of results for some proprietary models may be challenging due to potential API changes.

By providing these resources and detailed documentation, we aim to facilitate the reproduction of our results and encourage further research in this area. We welcome feedback from the community on any aspects that require additional clarification to ensure full reproducibility.

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

# A MORE EXPERIMENT RESULTS

Table 6: Results on Pure Textual Approximation for *Both Image and Audio*. All the images and audios are represented in texts. The results at the second and third column are taken from the corresponding models in Table 4 and Table 5.

| Input Context | Image Caption & Audio Transcript | Audio Transcript | Image Caption |
|---|---|---|---|
| LTU (7B) | 22.68% | 24.17% | 23.12% |
| Mu-LLaMA (7B) | 2.28% | 6.57% | 1.84% |
| MusiLIngo-long-v1 (7B) | 11.03% | 10.51% | 9.02% |
| Audio-SALMONN-13B | 36.95% | 34.41% | 33.36% |
| Qwen-Audio-Chat | 22.33% | 24.69% | 18.39% |
| Qwen2-Audio-7B-Instruct | 46.15% | 38.09% | 35.29% |
| Audio-Flamingo (1.3B) | 24.26% | 23.73% | 24.78% |
| InternVL-2-8B | 47.55% | 33.63% | 29.86% |
| InternVL-2-40B | 47.20% | 31.96% | 30.65% |
| Cambrian-13B | 43.08% | 31.96% | 29.16% |
| GPT4-o (0513) | 60.51% | 45.71% | 37.92% |
| GPT4-o (0806) | 53.77% | 47.55% | 29.51% |
| GPT4-o-mini | 51.05% | 49.04% | 32.84% |
| Gemini-1.5-Pro | 42.03% | 22.50% | 21.02% |
| Claude-3.5-Sonnet | 56.83% | 33.54% | 39.05% |
| Reka-core-20240501 | 42.23% | 36.33% | 32.94% |
| GPT-4V-Preview | 33.27% | 41.24% | 20.32% |
| GPT-4V-0409 | 29.95% | 45.80% | 20.84% |
| UnifiedIO2-large (1.1B) | 30.74% | 31.96% | 29.33% |
| UnifiedIO2-xlarge (3.2B) | 33.80% | 34.50% | 30.21% |
| UnifiedIO2-xxlarge (6.8B) | 34.15% | 29.77% | 27.15% |

# B DATASET DEVELOPMENT

## B.1 STATISTICS FOR OMNIINSTRUCT DATASET

Table 7: The Statistics of Data Filtering in OmniInstruct. The table shows the number changes of question-answer pairs before and after filtering from each of the data sources.

| Source | Original Train | Original Valid | Remained Train | Remained Valid |
|---|---|---|---|---|
| AVQA | 40,182 | 16,798 | 4,491 (11.2%) | 1,911 (11.4%) |
| Music-AVQA2.0 | 42,470 | 0 | 11 (0.03%) | 0 |
| MSRVTT-QA | 140,554 | 11,143 | 80,078 (57.0%) | 6,479 (58.1%) |
| Total | 233,206 | 27,941 | **84,580** | **8,390** |

As demonstrate in Table 7, most of the samples in the dataset are in low quality and therefore abandoned, and only 93k of samples remain for Omni-modality SFT training. This is reasonable because most of the questions are generated from templates, and the image may not sampled from the most relevant part of the questions and, therefore hot high in quality.

## B.2 PROMPT FOR QUALITY CONTROL ON OMNIINSTRUCT DATASET

## B.3 DIVERSITY OF MUSIC AUDIOS OF OMNIBENCHMARK

The music subset of our benchmark reflects a rich diversity of musical traditions, spanning a wide range of genres, styles, and cultural contexts. It encompasses Western classical symphonies, jazz chamber music, and avant-garde compositions alongside popular music from China, England, and France. Traditional forms like Kunqu opera and modern experimental pieces are represented, as well as instrumental music from regions such as India, the Arab world, Africa, and Japan. The benchmark also includes famous film soundtracks with various thematic elements and Asian folk oral traditions, such as chanting, drumming, and Humai. This eclectic collection, enriched by unique instances like famous concert spoofs and iconic YouTube parodies, ensures that each question offers a distinct challenge, showcasing the nuanced intricacies and breadth of global music heritage.

```
Initial Q&A: {question and answer}

The given Q&A is originally designed to answer based on the complementary context
built from an audio and an image together. Please evaluate whether the provided
Q&A is a bad/flawed sample due to one of the following reasons:

1. The answer could be inferred solely from the given image without the assistance
of audio;

2. The Q&A is not relevant to the image;

3. The Q&A is logically inconsistent.

After your evaluation, respond with 'Yes' if the Q&A is a flawed sample should be
removed, else response with 'No'.
```

Figure 6: The Prompt for OmniInstruct Dataset Filtering.

