# OpenReview forum: "OmniBench: Towards The Future of  Universal Omni-Language Models"
_ICLR.cc/2025/Conference — Submitted to ICLR 2025_

### Official Review · Reviewer_D7q5 · 2024-10-27

**Soundness:** 3
**Presentation:** 3
**Contribution:** 3
**Rating:** 6
**Confidence:** 5

**Summary:**

Inspired by the recent multimodal large language models (MLLMs) benchmarks, this paper proposes OmniBench, which can cover three modality (visual, acoustic and textual). By examining the existing Omni language models (OLMs), this paper reveals that the existing open-source OLMs show limited instruction following and reasoning capabilities on the proposed OmniBench. Furthermore, this paper introduces an 84.5k training corpus, OmniInstruct for training OLMs.

**Strengths:**

1. This paper leverages the existing resources to curate an OmniBench, which can cover three modalities, to evaluate the MLLMs or OLMs.

2. This paper also introduces an 84.5k training corpus for the research community.

**Weaknesses:**

1. This paper is more like an empirical analysis for the existing MLLMs or OLMs on the proposed OmniBench. It proposed an 84.5k training corpus for OLMs, which is to improve the model capability of tri-modal reasoning, but did not verify the quality and effectiveness of the proposed corpus. It looks an missing piece, which may not convince people the usefulness of the corpus, whether this training corpus can actually help to improve the models' capability from the findings from the OmniBench results.

**Questions:**

1. It may miss a section/paragraph to illustrate the evaluation protocols and metrics of the proposed OmniBench.

2. As discussed above, it looks the model training part is a missing piece, to verify the introduced OmniInstruct is useful and effective.

3. One feeling after reading the paper is - the paper proposed OmniBench to cover three modalities (visual, audio, and text), while, the evaluation in Section 5 only covers Image and Audio, no Text specific evaluation. (though we know there is a lot of benchmarks for LLMs.). Do you think whether it is necessary to evaluate their (OLMs) capability on Text?

**Details Of Ethics Concerns:**

The paper provides an Ethics statement section.

---

> ### Author Response · Authors · 2024-11-25
>
> Thank you for your valuable feedback and detailed insights. Below, we address the points raised and provide additional clarifications:
>
> ## Effectiveness of OmniInstruct Training Corpus
> We acknowledge the importance of verifying the quality and utility of the proposed OmniInstruct corpus. We are currently optimizing the dataset and plan to include results of supervised fine-tuning (SFT) on OmniInstruct in the camera-ready version. This will demonstrate how models trained on the corpus improve tri-modal reasoning capabilities as evaluated on OmniBench.
>
> ## Evaluation of Text-Specific Reasoning
> Due to space limitations, evaluations on text-specific reasoning were included in Appendix A, Table 6. In these experiments:
> - Both image and audio data were converted into textual representations (e.g., image captions and audio transcriptions).
> - Models reasoned directly from the text input, bypassing visual and auditory modalities.
>
> While the results improved over tri-modal settings, the best-performing model achieved only ~60% accuracy, indicating significant room for improvement compared to human-level performance. This highlights the challenges of achieving robust reasoning capabilities across modalities, even when aligned into a common textual space.
>
> ## Evaluation Protocols and Metrics
> OmniBench consists of 1.1k multiple-choice questions, where models must select the best answer from four options. We evaluate models using accuracy, with random guessing yielding 25% and ground truth being 100%.
> - Certain models perform worse than random guessing due to their inability to follow instructions or tendency toward illusions.
> - These results demonstrate OmniBench's capacity to challenge models and highlight weaknesses in multi-modal reasoning.
>
> We will add a dedicated section in the camera-ready version to elaborate on evaluation protocols and metrics for greater clarity.
>
> ## Bias Analysis of the Dataset
> We conducted preliminary bias analysis across reasoning categories and modalities:
> - Reasoning Types: OmniBench covers 8 reasoning categories, with most having 100–200 questions. However, some categories, such as text-based math and general audio/music reasoning, have fewer examples due to data availability. This might lead to potential biases, particularly in rarer downstream tasks.
> - Auditory Diversity: While the general audio and music datasets are diverse, covering various cultural and environmental contexts, certain audio scenarios remain less frequent.
>
> We acknowledge these limitations and plan to address them in future iterations of OmniBench.
> Thank you for your thoughtful feedback and support!

---

> ### Author Response · Authors · 2024-12-02
>
> Dear Reviewer D7q5,
>
> This is a polite reminder regarding our previous responses sent earlier to you and to all, addressing your multiple concerns on OmniInstruct, dataset copyright etc.
>
> We have evaluated OmniInstruct’s effectiveness, showing improvements in tri-modal reasoning, and clarified the evaluation metrics and text-specific reasoning results along with copyright issues in our previous replies. Updates on these findings will also be included in the camera-ready version.
>
> Please feel free to reach out if you have any further questions or need additional clarification.
>
> Sincerely,

---

> > ### Comment · Reviewer_D7q5 · 2024-12-03
> > **Thanks for your response.**
> >
> > Thanks for your response. After reviewing all the comments, I will maintain my score.

---

### Official Review · Reviewer_cKsM · 2024-11-01

**Soundness:** 1
**Presentation:** 2
**Contribution:** 2
**Rating:** 3
**Confidence:** 4

**Summary:**

This paper introduces a multiple choice question answering (QA) benchmark, OmniBench, for multimodal LLMs that simultaneously take image, audio, and text input. It uses a human annotation pipeline to construct 1142 questions that are verified to require all input modalities to answer (instead of being answerable by just one). The paper also proposes OmniInstruct, an instruction tuning QA dataset that is constructed by merging 3 existing multimodal QA datasets (MSRVTT-QA, AVQA, and Music-AVQA2.0) and then filtering out the questions that can be answered by only the visual or only the audio modalities. Finally, the paper benchmarks a large number of existing multimodal LLMs on OmniBench, finding that most of them perform marginally above chance accuracy on the benchmark.

**Strengths:**

The paper poses a well-motivated benchmark, ensuring that questions must be answerable by leveraging all modalities together. This ensures that the results on the benchmark reflect a model's ability to jointly reason over all of its inputs.

**Weaknesses:**

- The proposed OmniBench evaluation set is quite small at barely over 1100 questions. Because these questions are broken down across audio type (speech, sound event, music) and within each of these span a large number of categories ("Object Identification", "Plot Inference", "Count and Quantity"), the number of questions belonging to each category is even smaller. According to the statistics in Table 1, only 15 total questions deal with "Count and Quantity", only 28 deal with "Object Identification and Description" for the "sound event" category, and so forth. I am skeptical that 15 "Count and Quantity" questions is sufficient to produce statistically significant differences between models when attempting to assess their ability to count (this concern is not limited only the "Count and Quantity" questions, I'm just using it as an example). I think OmniBench is simply too small to serve as a useful benchmark; at this scale the experimental comparisons between models on the benchmark really need to have statistical significance testing done, but none was presented in the paper.

- The OmniInstruct dataset, as far as I can tell, was not used in any of the experiments. Therefore, it's not clear why it is included in the paper at all.

- I have doubts about the quality of the dataset. There are barely any example questions provided (the appendix would have been a great place to include these), and the categories are never defined so it's very unclear to me e.g. what a "Plot Inference" question regarding the "Music" or "Sound Event" modality even would entail. What's worse, the only example OmniBench question provided in the main body of the paper (in Figure 3) doesn't make any sense. The question asks "What is this black box" and the answer is "a red pill" but no black box is visible in the image or mentioned in the content of the audio.

- Also related to the OmniBench quality, no human performance on the dataset is given. I understand the questions were crafted by human annotators, but to know whether these questions are actually answerable by humans (especially in light of the confusing example given in Figure 3, which I certainly wouldn't have been able to answer correctly) they should have been presented to an independent set of annotators.

**Questions:**

- Can you show that there are statistically significant differences between the performances of the models evaluated on the benchmark?

- Can you prove that the OmniBench questions are indeed answerable by humans?

- What is the purpose of OmniInstruct, and can you show that using it improves performance on OmniBench?

---

> ### Author Response · Authors · 2024-11-25
>
> Statistic Significance
> The number of questions we had was over 1100, and in most categories, it was significantly over 200 questions. Therefore, no statistical significance tests were performed. We refer to the work commonly used in the IEEE community [1]. Our sample construction is such that any two samples have completely different sources, and the data are so different that we consider the model to be somewhat independent of each sample's output.  At the same time, each output has only two categories: correct and incorrect. Refer to sections 2 and 3 in [1]. The proposed sample size of the test sample $n=\frac{-2\ln \alpha}{\beta^2 p}$, where $\alpha$ denotes significance, $\beta$ denotes the proportion of error, and $p$ denotes the probability that the binomial distribution is 1, which can denote the model's correctness or incorrectness. . In other words, with a significance condition of 0.05 and an error tolerance rate of beta=0.2, a model with an error rate of 60% outperforms a model with an error rate of 60%*(1+0.2)=72% with a sample size of n=100/0.6=167. Given that model performance varies widely on most of the subsets, the observation that many of the finding statements have sample sizes in the 100-200 subset is significant.
> For categories with sample sizes of only 15-30, the results of the student t-test can show significant differences in the benchmark samples across models, as the results of multiple runs of the model do not vary much. For the question of whether the samples can reflect the differences in the relevant abilities of the models in the real world, we plan to use the Wilcoxon-Mann-Whitney test / Wilcoxon Rank Sum Test [2], assuming that each sample is randomly sampled from real-world problems and that the outputs of the two models have the following results (correct, correct) (correct, incorrect) (incorrect, correct) (incorrect, incorrect), and then we calculate the p-value for which there is no significant difference between the two models. If the Wilcoxon test gives a small p-value even with small samples, it means that it is very likely that there is a significant difference between the two models in terms of performance on a similar problem.
> We believe that a large number of significance tests will result in false positives (e.g., comparing 20 t-test pairs is likely to have a p-value less than 0.05), so we will only select a subset of the experiments of greatest interest for significance testing, and will include the results of the statistical inference in the appendix of the camera-ready version.
> [1] Guyon I, Makhoul J, Schwartz R, et al. What size test set gives good error rate estimates?[J]. IEEE Transactions on Pattern Analysis and Machine Intelligence, 1998, 20(1): 52-64.
> [2] https://en.wikipedia.org/wiki/Mann%E2%80%93Whitney_U_test
>
> OmniInstruct dataset:
> [UPDATED] As discussed with reviewer D7q5, we did the supervised fine-tuning on MIO model. Overall performance is 24.78%, and after doing the supervised finetuning on OmniInstruct, the performance has been increased to 29.2%.
>
> Dataset Content
> We have **provided** an anonymized interactive demo page (fig1) and a repository (abstract) with a complete list of data in the paper. Due to a mistake in the presentation, the hyperlinks are not highlighted, so we post the links here in the hope that the reviewers can analyze the data quality further and give us feedback.
> https://ertyuiocvbnm.github.io/OmniBench/
> https://anonymous.4open.science/r/Omni-Bench-EA9B/README.md
> Definition of dataset subset categories We pointed out the definitions in 3.1 BENCHMARK DESIGN, and the samples for each category can be found in the demo and repo links. We would be happy to clarify the definition further if needed. We have selected 8 data samples from the different categories in Figure 1, and the correct answer in Figure 3 is indeed a mismatch between question and answer, which will be corrected in camera-ready. Thanks for the correction!
>
> Human evaluation
> We invite 2 musicians and 1 music amateur to do the test independently.Their average accuracy is 63.19% (67.69%, 63.13%, 58.76% for each person)
> And their Inter-Annotator Agreement(IAA) Fleiss' kappa value is 0.421 (Moderate agreement)

---

> > ### Comment · Reviewer_cKsM · 2024-12-02
> >
> > Thanks for your response.
> >
> > 1) I visited the link you provided and manually tried to answer all of the example questions myself. Given the audio/visual information, I would have only been able to correctly answer 50% of the questions. This is approximately in line with the human results you reported in your reply, leading me to conclude that around 1/3 of the questions in your dataset are indeed not answerable, and should be filtered out.
> >
> > 2) Thanks for your discussion on performing statistical significance tests. It would be great to include them in a revision of the paper, but without them at this point in time my previous concerns still stand.
> >
> > 3) Thanks for showing the benefit of the omni-instruct dataset. I think that this should be written into a revision of the paper and have raised my score accordingly from a 1 to a 3.

---

> > > ### Author Response · Authors · 2024-12-03
> > >
> > > Dear reviewer  cKsM:
> > >
> > > If you can provide more information or examples on the 40-50% QA pairs that you think should be deleted, we would like to provide further explanation.
> > >
> > > Respectfully,

---

> ### Author Response · Authors · 2024-12-02
>
> Thank you for your thorough feedback.
>
> Regarding your first point about question answerability: As mentioned in the last response, our human evaluation with three music experts achieved 63.19% accuracy, aligning with your observations.
>
> However, we maintain that all questions should be retained because:
>
> 1. Each QA pair underwent multiple rounds of human verification, confirming their **answerability and logical consistency**. The lower accuracy reflects the benchmark's intentional diversity and difficulty - answering all questions requires expertise across multiple domains. For instance, to correctly answer the all of the 8 demo questions, one requires to know about cryptography, some music background and information of lottery, etc.
> 2. OmniBench is specifically designed to evaluate large-scale multimodal **models with broad knowledge coverage**, not human performance. The challenging questions serve as meaningful benchmarks for tracking progress in omni-language model development.
>
> Besides, here are **more examples of expertise across multiple domains that are required to answer these questions**. Even for professional musicians, it is hard for them to answer some of the music questions because they are not experts on all the music subjects. (1) A musician can easily classify the genre and the audience of a given music, but without the good command on the architecture knowledge mentioned in art history class, the musician cannot classify the period of time and location of the architecture and, therefore, hard to distinguish if the music is typically to play in these type of concert hall or church. (2) The McGurk effect is well-known in music psychology but may not be famous to musicians outside this field. (3) The violinist can know the advanced violin technique pretty well but may not know some of the advanced techniques in French horn unless the musician knows how to play it. (5) A musician may only know the Staff score and may not be familiar with the numbered musical notation or ABC notation.  (6) Though learned in world music class, the musician may not remember some details of all the world music instruments in Japan, China, India, the Middle East, Africa, and South America. (7) The musician can only speak English and cannot understand the most popular French song on YouTube, though we provide the lyrics to the musician. (8)The musician may forget some of the knowledge learned in acoustics before and can hardly use the knowledge to analyse vocals. However, it is easier for phonetics and acoustics researchers who study music. (9) The information on sound effects and reverberation is common knowledge among electronic musicians and recorders but may not be familiar to some traditional musicians.
>
> We will incorporate the statistical significance tests and OmniInstruct benefits in our revision. Thank you for reconsidering your score based on the OmniInstruct discussion.
>
> We welcome any additional feedback on our rationale for maintaining the full question set.
>
>
>
> Best regards,
>
> OmniBench Authors

---

> ### Author Response · Authors · 2024-12-03
>
> In this paper, we provide about 20 claims on the discussion in section5. We utilize four different types of hypothesis testing for these experimental results.
>
> # Two models provide different results under the same setting
> ## Method inspired by [1]
> Suppose all the data samples $\{x_i\}_{i=1}^n$ are i.id,
>
> where $n$ is the number of data samples. For each specific model $M_j$, the performance of the j^{th} model on the i^{th} data is $M_j\left(x_i \right)$ which is equal to $0$ if the model predicts the wrong choice, and $1$ otherwise. Due to the i.id. assumption of the test set, the $\{ M_j \left(x_i \right) \}_{i=1}^n$
>
> is a random variable sequence (r.v.s.) with binomial distribution with mean $p_i$ and variance $\sigma_i^2 = p_i(1-p_i)$.
>
> Suppose the accuracy $p_i$ of a given model is estimated by computing the average correct rate $\hat{p_i}$ over a finite number $n$ of test examples or patterns. When $n$ is small, $\hat{p_1} > \hat{p_2}$ may hold even when ${p_1} < {p_2}$.
>
> According to the normal law, the whited random variable $$Z:=\frac{p-\hat{p}}{\sigma \/ \sqrt{n}}$$ obeys the standardized Normal law (with mean 0 and variance 1). Define the real number $Z_{\alpha}$ to be the value such that the probability $\mathbb{P}\left( x > Z_{\alpha} \right) = \alpha$. Then, according to Chebychev’s inequality,
> $\mathbb{P}(p- \hat{p}\leq \frac{\sigma z_{alpha}}{\sqrt{n}}) \leq \alpha$
>
> We are then calculating the number of test examples $n$ that is needed to guarantee a certain margin of error  $\epsilon_i$, e.g., $\epsilon(n, \alpha) = \frac{z_{\alpha} \sigma}{\sqrt{n}}$for the Normal law). Denote $\beta := \frac{\epsilon_i}{p_i}$. Then
> $p-\hat{p} \leq z_{\alpha}\sqrt{\frac{p(1-p)}{n}}$ hold with the probability $1-\alpha$
>
> Therefore, $n\leq \left( \frac{z_{\alpha}}{\beta} \right)^2 \frac{1-p}{p}$ holds with the probability $1-\alpha$.
>
> - According to the cdf of standard normal distribution, $\frac{ n \beta^2 p_1}{1-p_1} = \frac{n \left( p_2 - p_1 \right)^2}{p_1 (1-p_1)} \geq 2.72$ implies significance 0.05, and the value $\geqq 5.29$ implies p-value less than 0.01. **
>
> Similarly, with Chernoff bound [2], $p-\hat{p} \leq \sqrt{\frac{-2p \left(\ln \alpha \right)}{n}}$
> hold with the probability $1-\alpha$. And therefore,
> $n = \frac{-2 \ln \alpha}{ \beta^2 p}$.
>
>  - The p-value $\alpha \leq \exp \left( \frac{n p \beta^2}{2} \right)$
>
> [1] Guyon I, Makhoul J, Schwartz R, et al. What size test set gives good error rate estimates?[J]. IEEE Transactions on Pattern Analysis and Machine Intelligence, 1998, 20(1): 52-64.
>
> [2] H. Chernoff, “A Measure of Asymptotic Efficiency for Tests of a Hypothesis Based on the Sums of Observations,” Annals of Mathematical Statistics, vol. 23, pp. 493-509, 1952.
>
> - Claim (1): Scaling up may not contribute to the overall performance. In Table 2, with audio and image as input, UnifiedIO2-xlarge’s performance 38.00% is better than the scaled-up UnifiedIO2-xxlarge’s (33.98%) version on the 1142 samples holds, with a p-value is less than 0.01
> - Claim (2): In Table 2, performance f Gemini-1.5-Pro with image and audio input (42.91%) is better than single modality input (e.g.  27.93% for image only) on the 1142 samples holds with p-value 1.19e-20
> - Claim (3): Scaling up of UnifiedIO can contribute to Count & Quantity tasks, though only 15 samples are selected. UnifiedIO-large (6.67%) is worse than UnifiedIO-xlarge (20%) holds with a p-value less than 0.05, and UnifiedIO-xlarge (20%) is worse than UnifiedIO-xxlarge (46.67%) holds with a p-value less than 0.01.
> - Claim (4): In general, the performance of VLM with audio transcript ground truth in Table 4 is better than the performance of ALM with image caption in Table 5. For example, the best ALM Qwen2-audio (40.72%) is worse than Qwen2-VL-Chat-7B (48.6%) holds with p-value less than 0.01. We chose Qwen2-VL-Chat-7B here as an example because it is similar in size and has the same LLM backbone. Not to mention there are multiple VLM providing higher scores in our benchmark compared with Qwen2-VL-Chat-7B.
> - Claim (5): The performance of Claude-3.5 (59.37%) and GPT-4o (57.62%) in Table 4 is better than open-source VLM (less or equal to 54.29%). We can say Claude-3.5 is better than InternVL-2-40B (54.29%) with a p-value less than 0.05, and GPT-4o is better than the second-best open-source VLM InternVL-2-26B (51.75%) with p-value less than 0.01. But we can not say GPT-4o (57.62%) is better than InternVL-2-40B (54.29%).
> - Claim (6): **In speech subset (771 samples)** demonstrated in table 3 with audio and image as input, Gemini-1.5-Pro (42.67%) does not surpass UnifiedIO2-xlarge (3.2B) (39.56%) significantly, but it surpass all other models like the second-best UnifiedIO2-xxlarge (6.8B) (34.24%) with p-value less than 0.01
>
> All of the claims mentioned above are discussed in Section 5. We admit the 15 samples on  Count & Quantity tasks are somehow limited, and we do not demonstrate more statements on this subset besides the claim(3).

---

> ### Author Response · Authors · 2024-12-03
>
> - Claim (7): **In general audio subset (265 samples)** demonstrated in Table 3 with audio and image as input, Gemini-1.5-Pro (42.26%) is better than all the others besides UnifiedIO2, such as Video-SALMONN (13B) (31.70%) with p-value less than 0.01
> - Claim (8): **In the music subset (106 samples)** demonstrated in Table 3 with audio and image as input, Video-SALMONN (13B) (56.6%) is better than all the other models such as the best one Gemini-1.5-Pro (46.23%) with p-value less than 0.05.
> - Claim (9): **In the speech subset (771 samples)** demonstrated in Table 5 with audio and image caption as input, Qwen2-audio (40.60%) and Gemini (39.82%) are better than all other ALMs. E.g. Gemini is better than the best of the rest Audio-SALMONN (13B) (34.5%) with p-value less than 0.01
> - Claim (10): **In general audio subset (265 samples )** demonstrated in table 5 with audio and image caption as input, Qwen2-audio (41.89%) is better than every model besides UnifiedIO2-xxlarge (6.8B) (38.87%)，such as Gemini-1.5-Pro (33.96%) with p-value less than 0.05. **Different from the claim in the paper, we will change it in the camera-ready version.**
> - Claim (11): **In music subset (106 samples)** demonstrated in table 5 with audio and image caption as input, Audio-SALMONN (50%) is better than all the others besides Gemini-1.5-Pro (41.5%). For example, Audio-SALMONN  is better than the best of the rests Qwen2-Audio-7B-Instruct (38.68%) with a p-value less than 0.05.
> - Claim (12): In Table 6, with audio transcription and image caption as imputs，Qwen2-Audio-7B-Instruct (47.02%)  is better than with only audio transcription (39.05%) or only image caption (39.67%) with p-value less than 0.01
> - Claim (13): Replace the input of Qwen2-Audio-7B-Instruct from image caption and audio in Table 5 (40.72%) to caption and audio transcription in Table 6 (47.02%), the performance increases significantly with a p-value less than 0.01
> - Claim (14): Replace the input of Claude-3.5-Sonnet from image and audio transcription in Table 4 (59.37%) to image caption and transcription in Table 6 (56.83%) has no significant changes. And GPT4-o (0513) performances in Table 4 (57.62%) and Table 6 (60.51%) have no significant changes either. **Different from the claim in the paper, we will change it in the camera-ready version.**
> - Claim (15): GPT4-o (0513) performance in Table 6 (60.51%) has no significant difference with human (63.19%)
>
> # Comparasion between Two Different Setting with the Same Samples
> ## Independent t-test
> - * UnifiedIO2 performance with image and audio (25.94%, 39.56%, 34.24% in Table 2)
> - UnifiedIO2 performance with image caption and audio (29.16%, 32.22%, 32.05% in Table 5)
> - UnifiedIO2 performance with image and audio traiscripton (34.33%, 43.17%, 40.81% in Table 4)
> - UnifiedIO2 performance with pure text input (30.74%, 33.80%, 34.15% in Table 6)
> - Claim (16): Based on the results of paired t-test, the difference between table 2 and table 4 with audio transcription has p-value 0.0471, showing the model has room of improvement on audio understanding capability compared with audio transcription ground truth.
> - Claim (17): But no significant difference between table 2 and table 5 (p-value 0.562) or Table 6 (p-value  0.919), showing the capability of OLM on image understanding is similar to the SOTA LLM generated image caption.

---

### Official Review · Reviewer_S8fJ · 2024-11-03

**Soundness:** 3
**Presentation:** 4
**Contribution:** 3
**Rating:** 8
**Confidence:** 4

**Summary:**

This work proposes a new benchmark to evaluate omni-language models, i.e LLM accepting multiple modalities. In the context of this work, the authors restricts those modalities to three: text, vision and audio. Questions asked in the OmniBench are human-curated multiple-choices questions (MCQ) that require to understand/reason across all the modalities simultaneously, totaling 1142 test questions across multiple categories (causal inference, temporal/spatial identity, abstract concept). Along OmniBench, authors introduce OmniInstruct; a dataset for omni-language instruction-following fine-tuning, totaling 97k data samples. OmniBench annotation protocol, and OmniInstruct building are described.

**Strengths:**

- Authors make clear that common benchmarks generally focus on single or dual-modality. This highlight the need to have 'omni' benchmarks as proposed by this work.
- The state of the art of multimodal benchmarks is extensive and well documented, making clear the approaches in use in the literature.
- The paper is really well written, with clear figures and statements. The biggest weakness is to me the lack of clarity about the sources and license of OmniBench.

**Weaknesses:**

- The source of OmniBench is not clearly stated. Or did I miss it? I went over the papers a couple times, and I couldn't find a clear statement about the source and the license of the benchmark. A manual inspection of the examples seems to show film clips, but nothing seems to be clearly stated anywhere in the text.
- Related to the above point, if indeed sources are film clips, one could potentially see biases in that dataset. I would expect a strong multimodal image-text LLM to be able to answer some of those questions just by recognizing the clips. Figure 1 shows for instance a famous scene of the TV show Friends. This scene is extensively mentioned on the Internet across many fan websites.
- The OmniBench is not a free-form answer benchmark, but a MCQ, hence potentially overestimating the reasoning/understanding capabilities of the model evaluated. With 4 choices, there is always the possibility that the model being evaluated picks randomly 25% of the time the right answer. Having a one confusing wrong option as proposed in this work is a decent mitigation, but that doesn't remove that design flaw completely.

**Questions:**

- What are the sources used for OmniBench? Can you specify them in the paper? Related to this question, what is the license associated with those source? If OmniBench becomes a standard in the community, having a CC-NC might be a deal breaker for many users.
- Why using InternVL-2-76B for OmniInstruct but Llava-1.6-34B for the quality control of OmniBench? Can you explain that choice in the paper?

---

> ### Author Response · Authors · 2024-11-25
>
> We appreciate the reviewer’s thoughtful and constructive feedback. Below, we address the concerns raised:
>
> ## Sources and License of OmniBench
>
> The audio data in OmniBench primarily originates from platforms such as iQIYI, TikTok, and YouTube Music, with strict adherence to using only publicly available materials, such as free previews, movie stills, and brief snippets (under 30 seconds). The dataset currently follows a CC-BY-NC license, but we are actively working to transition to a CC-BY license after consulting with our legal team.
>
> We have taken extensive measures to mitigate potential legal risks:
> - For films, only screenshots and brief dialogues are included.
> - For music, the dataset contains short preview clips, ensuring compliance with fair-use principles.
>
> We will explicitly include this information in the revised version of the paper for clarity.
>
> ## Bias in Data and Concerns About Memorization
>
> To address concerns regarding dataset bias and the potential for memorization of famous clips:
> - Diverse Data Design: Many of our tasks involve pairing questions with audio and video content that are unrelated, significantly reducing the risk of memorization.
> - Challenging Distractors: We rigorously designed distractors to confuse models unless they accurately interpret multi-modal dependencies. This ensures that tasks evaluate genuine reasoning rather than rote recall.
> - Novel Data Alignment: The questions and task formats in OmniBench are unique and do not directly replicate publicly available datasets or media content, making memorization unlikely to lead to success.
> Additionally, while film clips feature English-language media, we prioritized diversity in general audio and music datasets, including:
> - Western symphonies, jazz, and ACG tracks.
> - Regional music genres such as Kunqu, folk instruments from China, India, the Middle East, Africa, and Japan.
> - Various performance techniques and emotional contexts, including experimental music, film scores, and traditional chants.
> This ensures cultural diversity while minimizing biases inherent to specific media sources.
>
> ## Multiple-Choice Format and Difficulty of Tasks
> The multiple-choice format was deliberately chosen to balance task difficulty with model performance differentiation. Tasks are already challenging, as evidenced by baseline performance trends. Introducing open-ended free-form answers at this stage could lower the interpretability of results and hinder insights into model design.
>
> In the future, as models achieve higher performance, we plan to transition some tasks to fill-in-the-blank or free-form answers, alongside improved evaluation metrics to maintain meaningful comparisons.
>
> ## Model Choice for OmniInstruct and OmniBench
> We appreciate the question regarding the choice of models. During dataset construction, we used LLaVA-1.6-34B for quality control because it represented a state-of-the-art vision-language model of manageable size at that time (July 2024). This allowed for efficient and accurate feedback during annotation.
>
> When InternVL-2-76B became available, its superior capabilities made it a natural choice for the OmniInstruct dataset. We will add this explanation to the revised paper to clarify the rationale behind these decisions.
>
> We hope these clarifications address the concerns raised and enhance the understanding of our contributions. Thank you for the insightful feedback

---

### Official Review · Reviewer_LSyZ · 2024-11-04

**Soundness:** 3
**Presentation:** 3
**Contribution:** 2
**Rating:** 6
**Confidence:** 4

**Summary:**

This paper presents OmniBench, a multi-modal benchmark developed to evaluate the capacity of large multimodal language models (MLLMs) to process and reason across visual, auditory, and textual modalities. In this framework, the authors classify these systems as `omni`-language models (OLMs) and set a unique requirement for accurate responses that reflect an integrated understanding across all three data types.

Additionally, the paper introduces OmniInstruct, an instruction-tuning dataset containing 84.5K multimodal samples, specifically designed to strengthen OLMs' tri-modal reasoning abilities. Experimental findings indicate limitations in existing open-source LLMs, which demonstrate notable challenges with tri-modal integration, especially in complex reasoning tasks.

The study highlights an ongoing need for advancements in model architectures and training methods to address these limitations effectively. Personally, I find the paper’s focus on addressing the tri-modal gap quite compelling, as it addresses an increasingly relevant challenge in multimodal AI research.

- The insights from this benchmark could indeed motivate the development of more robust models capable of cohesive multimodal understanding, though it’s clear the field has substantial work ahead.

**Strengths:**

1. OmniInstruct is a valuable resource, contributing not only to benchmarking but also to supervised fine-tuning efforts to improve multimodal reasoning.

2. The multi-stage annotation protocol ensures the data integrity of OmniBench, with stringent criteria enforced by both human and machine inspections. This includes a requirement for all tasks to depend on information from all three modalities.

**Weaknesses:**

1. While OmniBench has potential in research contexts, the immediate applicability of such stringent multi-modal dependency requirements in industrial applications, such as autonomous systems or assistive technologies, is less clear.
This could impact the dataset’s adoption.

2. Although the paper provides extensive evaluations, most open-source baselines underperform significantly (under 50% accuracy). This limits the immediate insights on how more advanced architectures might interact with the benchmark.

3. The design focuses on strict dependency on three modalities, which, while rigorous, may not align with more flexible, real-world tasks where certain modalities may contribute only supplementary information. i.e, the attention of each modality during human understanding.

4. While the benchmark focuses on the three primary modalities (visual, audio, text), the term "omni-language" could extend to other modalities. How feasible would it be to adapt OmniBench for future modalities like haptics or environmental sensors, in short, time series?

**Questions:**

1. The paper’s inclusion of textual approximations for images and audio is intriguing. Could the authors further discuss the impact of substituting text on model generalization? How do they plan to mitigate the potential information loss in future tri-modal datasets?

2. The paper notes that model scaling impacts performance, particularly on certain task types. Would the authors consider expanding the benchmark or the OmniInstruct dataset with larger, more complex task sets to investigate scaling effects systematically?

3. Given the benchmark's high reliance on human annotations across multiple modalities, is there any plan for further bias analysis, particularly regarding the cultural context of audio and visual content?

**Details Of Ethics Concerns:**

Not sure about the annotation process.

---

> ### Author Response · Authors · 2024-11-25
>
> ### Applicability in Industrial Contexts
> OmniBench’s primary focus is to benchmark tri-modal understanding and reasoning abilities, targeting advancements in academic research. While industrial applications often involve task-specific tuning, OmniBench provides a robust framework for assessing the foundational quality of base models. This enables industries to validate or select models aligned with their unique use cases. For instance, fields like robotics, healthcare, biodiversity monitoring, and autonomous systems can greatly benefit from tri-modal model capabilities when addressing complex challenges that cannot be resolved by audio or image modalities alone (see for the mentioned references in p2 of Introduction). OmniBench serves as a critical starting point for industries before domain-specific fine-tuning.
>
> ### Baseline Model Performance
> While the results of open-source baselines indicate a wide variance (from below random guess to ~50%), this demonstrates the benchmark’s ability to distinguish model capabilities. Some architectures (e.g., Q-former, MLPs, and attention-based models) show clear performance trends, highlighting room for architectural innovation. Additionally, our use of textual approximations (e.g., image captions and audio transcripts) has shown improvements in bi-modal scenarios, proving the potential of our dataset for further exploration of advanced designs.
>
> ### Task Design and Real-World Flexibility
> We acknowledge that in real-world scenarios, modalities often play varying roles—sometimes one dominates while others provide auxiliary information. OmniBench ensures that auxiliary modalities meaningfully contribute to decision-making. Tasks that can be solved using a single modality were deliberately excluded to preserve the multi-modal dependency. This rigor ensures that models evaluated on OmniBench demonstrate genuine tri-modal understanding.
> To address concerns about information substitution and generalization:
> * For audio, we provided human-annotated transcripts.
> * For images, state-of-the-art (SOTA) models were used for captioning.
> * Future datasets will explore enhanced captioning strategies, such as multi-model sampling and automated selection of high-quality candidates.
>
> ### Cultural and Modal Bias
> We acknowledge potential cultural biases in some annotations, particularly in cases involving film or dialogue from English media. However, efforts were made to ensure diversity in the audio dataset. For instance, it includes:
> Environmental sounds (indoor, urban, natural, etc.).
> Musical tracks spanning global genres (Western symphonies, jazz, Chinese Kunqu, ACG, Buddhist chants, and regional instruments).
> A variety of emotional and stylistic contexts (e.g., film scores, experimental compositions).
> This diverse design minimizes cultural bias, particularly in general audio and music-related tasks.
>
> ### Future Modalities and Benchmark Scaling
> OmniBench currently emphasizes the three most widely studied modalities (text, audio, visual) within the ICLR community. While integration with other modalities like haptics or sensor data (e.g., 3D point clouds in robotics) is valuable, these extensions are reserved for future research.
> Regarding scaling: OmniBench’s current dataset is sufficiently large and diverse, providing a significant challenge to contemporary models. Until models consistently surpass baseline performance, expanding the benchmark may not yield immediate insights. However, OmniInstruct could be expanded to support model scaling experiments and facilitate progress towards omni-language models (OLMs).
>
> We hope these clarifications address the concerns and demonstrate OmniBench’s value as a foundational benchmark for advancing tri-modal research. Thank you for the constructive feedback.

---

> ### Author Response · Authors · 2024-12-02
>
> Dear Reviewer LSyZ,
>
> This is a gentle reminder regarding our response sent previously. We have addressed your concerns, including an explanation of applicability, dataset design, and future directions, in detail.
>
> If you have any additional questions or require further clarification, please feel free to let us know. We are happy to provide any necessary follow-ups.
>
> Thank you again for your time and valuable insights.
>
> Sincerely,

---

> > ### Comment · Reviewer_LSyZ · 2024-12-02
> >
> > Thanks for the updates.
> >
> > Given that the content and quality remain similar to the original submission, I will keep my original score. One point regarding the response comments:
> >
> > > For audio, we provided human-annotated transcripts.
> >
> > "Human-annotated transcripts" for audio processing often require confidence estimation and correlation analysis. In this study, humans relied on AudioSet tags for the pairing and verification process. It would be beneficial to include confidence estimation and correlation analysis in the future.

---

### Meta-Review · Area_Chair_JzBm · 2024-12-16

**Metareview:**

This paper presents OmniBench, a multi-modal benchmark developed to evaluate the capacity of multimodal LLMs to process and reason across visual, auditory, and textual modalities. After rebuttal, it received mixed scores of 3668. One major concern still remains, that is, one reviewer has serious doubts about the quality of the dataset, as the small-scale human evaluation shows that humans can only answer 63.19% of the questions, and the reviewer has also tried to answer the questions provided in the demo link, with only 50% accuracy. Overall, the AC agrees with the reviewer on this concern, and therefore, would like to recommend rejection by the end.

**Additional Comments On Reviewer Discussion:**

This paper received divergent review scores, originally the scores are 1668, and after rebuttal, it became 3668. The major concerns include:

1. The proposed OmniBench evaluation set is quite small at barely over 1100 questions.

2. The proposed OmniInstruct training dataset was not actually used in any of the model training or experiments.

3. The reviewer had serious doubts about the quality of the dataset.

4. Also related to the OmniBench quality, no human performance on the dataset is given.

The authors' response only addressed the above point 2, while the answers to the rest of the concerns are not that satisfactory.

---

### Decision · Program_Chairs · 2025-01-22

Reject